# A Comprehensive Review on Utilization of Slaughterhouse By-Product: Current Status and Prospect

Derseh Yilie Limeneh [1,*], Tamrat Tesfaye [1,2], Million Ayele [1], Nuredin Muhammed Husien [1], Eyasu Ferede [1], Adane Haile [1], Wassie Mengie [1], Amare Abuhay [1], Gemeda Gebino Gelebo [1], Magdi Gibril [2] and Fangong Kong [2]

1   Biorefinery Research Center, Ethiopian Institute of Textile and Fashion Technology, Bahir Dar University, Bahir Dar P.O. Box 1037, Ethiopia; tamrat_tsfy@yahoo.com (T.T.); milliti2000@gmail.com (M.A.); babinure@gmail.com (N.M.H.); eyasuferede1982@gmail.com (E.F.); adaneh77@gmail.com (A.H.); wassie215@gmail.com (W.M.); amare2000et@gmail.com (A.A.); gemedalvgebino@gmail.com (G.G.G.)
2   State Key Laboratory of Biobased Material and Green Papermaking, Qilu University of Technology, Jinan 250316, China; magdi.gibril@gmail.com (M.G.); kfgwsj1566@163.com (F.K.)
*   Correspondence: derseh2003@gmail.com

**Abstract:** The meat processing industry produces a huge quantity of by-products, approximately 150 million tonnes per year. The live weight of the animals is distinguished as edible, inedible, and discardable by-products, with the discardable parts equating to 66%, 52%, and 80% of the overall live weight of cattle, lamb, and pigs, respectively. Only a small percentage of those by-products are nowadays exploited for the production of high added value products such as animal feed, glue, fertilizers, etc., whereas the main management method is direct disposal to landfills. As such, the current disposal methodologies of these by-products are problematic, contributing to environmental contamination, soil degradation, air pollution, and possible health problems. Nevertheless, these by-products are rich in collagen, keratin, and minerals, being thus promising sources of high-value materials such as bioenergy, biochemical and other biomaterials that could be exploited in various industrial applications. In this paper, the possible utilization of slaughterhouse by-products for the production of various high added value materials is discussed. In this context, the various processes presented provide solutions to more sustainable management of the slaughterhouse industry, contributing to the reduction of environmental degradation via soil and water pollution, the avoidance of space depletion due to landfills, and the development of a green economy.

**Keywords:** slaughterhouse by-product; collagen; keratin; biorefinery

## 1. Introduction

Great consideration has been given lately to the conversion of slaughterhouse by-products to usable biomaterials, instead of disposal via landfilling, incineration, and burial, to minimize environmental and health issues, in addition to mitigating the depletion of space due to landfills. To reduce environmental problems due to the landfilling of by-products, consideration of the production of fertilizer, animal feed, and biomaterials should be taken as effective utilization of slaughterhouse wastes. An excellent option for handling these environmental and health problems is using an integrated biorefinery technique for the conversion of slaughterhouse waste into biomaterial, biofuel, biogas, and biochemicals [1]. Hence, by following an integrated biorefinery approach, the by-products of the slaughterhouse can be utilized appropriately, not only to reduce pollution (contamination) but also to create job opportunities [2]. Slaughterhouse by-products are taken to be secondary products obtained during the slaughtering and processing of livestock to get meat and by-products exist in liquid, solid, and semi-liquid forms [3,4].

Solid management is a rising concern, not only for developing countries but also for the world as a whole [4]. Worldwide, slaughterhouse industries produce a large number of

organic by-products; however, conversion of by-products to usable biomaterials has great importance in facilitating a green economy and the consistent supply of biomaterials for its industries [5]. Slaughterhouse by-products are protein-rich biomass consisting of collagen (gelatine), keratin, fats, amino acids, and mineral products. The keratin and collagen parts are a valuable resource and their utilization results in consistent conversion into value-added materials, which leads to the product development of cost-effective biomaterials.

In the meat processing industry, by-products make up 66% of the live load of cattle, 80% of the live load of pigs, 62% of the live load of broilers, and 57% of the live load of fish, being considered inedible by humans [6]. The meat processing industry plays a valuable role in sustaining the production of livestock, the livelihood of rural areas, and the earning of foreign currency. Slaughterhouses not only provide meat but also by-products for the people [7,8].

Worldwide, meat consumption is increasing as income levels and population growth increase. The majority of studies on the possible valorization of slaughterhouse by-products deal with their conversion to animal feeds and fertilizers. The current work goes a step beyond this, reviewing possible alternative options for the utilization of by-products in various high-added-value biomaterials. Thus, the novelty of this review is in the valorization of slaughterhouse by-products as high-value products and, at the same time, reducing problems related to waste disposal [9]. Hence, instead of landfilling or incineration of by-products, their conversion into high-value products such as composite materials, re-generated fibers, biomedical tools (tissue engineering, scaffolds, drug delivery carriers), electronics (electronic packaging, wearable electronics, fuel cells), and chemicals for various industries [10,11]. These applications may potentially consume the majority of the huge quantities of by-products produced by the meat processing industry.

The review is tasked with responding to the priorities of action plans of the global strategy for sustainable development for optimization of biomass processing industries whilst ensuring economic growth with reduced environmental impact, i.e., resource-efficient, low carbon, and pre-employment growth paths [10]. Additionally, it aligns itself with a waste road map for organic by-product streams that support the maximization of the diversion of by-products from landfills towards value-adding opportunities [11,12]. The review shows the possibilities of utilization of abattoirs' by-products as high-value products [12,13].

## 2. Overview of the Slaughterhouse Processing Industry

The livestock sector and meat processing industries are the highest contributors to environmental degradation and their by-products amount to nearly 150 million tonnes per year [14,15]. The increasing demand for animal protein has led to the global livestock revolution, with significant implications for the environment and our health [16,17].

The annual slaughter capacity in Ethiopia reaches 4.6 million tonnes of beef, 1.5 million tonnes of poultry and 0.105 tonnes of fish and its by-product percentages vary from 40–60%, 10–45% and 25–70%, respectively [17,18]. In the meat processing industry, the majority of products consist of edible products and in-edible by-products [16]. Some of the by-products found in the slaughterhouse include tendons, skin, bones, gastrointestinal tract, blood and internal organs [11].

Proper utilization of slaughterhouse by-products has a positive significance for the meat industry. Its increment should be 11.4% from beef and 7.5% from pigs from abattoir by-products [11]. Most papers show the by-product content of cattle, lambs, and pork as 66%, 58% and 52%, respectively (organs, blood, bone, intestinal and abdominal contents, skin and fat). Almost 50% of meat processing animal by-products are not directly consumable; as a result, favorable sources of income are lost [19,20].

Most of the literature is focused on waste disposal systems; however, this review focuses on how by-products can be converted into valuable materials.

*Slaughterhouse by Product*

The yield percentage of the animal by-product can tremendously vary in terms of living topographic area of the animal, sex, load weight, method of collection, and fat percent of the animals [21] Detailed classifications of animal by-products are found in Table 1. The common disposal techniques used still are controlled landfilling (by-product dump to the landfill site), incineration (thermal destruction technology), and burial [22,23].

**Table 1.** Classification of slaughterhouse by-product and its utilization area [22].

| Slaughtered Animal | Classification | Utilization | | |
|---|---|---|---|---|
| Animal | Dressed carcass part (meat) | Used for human consumption | | |
| | Non-carcass part (by-products) | Edible | Red offal | Liver. kidney, heart. tongue, trachea, lung and spleen | Human consumption |
| | | | White offal | Stomach, intestine, Gizzard | |
| | | | Dark offal | Head, Neck, Trotter, shank | |
| | | Non-edible | Re-processed | Blood, hoof, horns, Gland, Bones, fats and skin/hide | Used in animal feed industry, Fabric/Cosmetic/ Pharmaceutical industry |
| | | | Discard | Gastrointestinal tract content, trimming, fetus | Used in biogas/fertilizer industry and animal feed industry |
| | Condiment part | Discard | | Gastrointestinal tract content, trimming, fetus | Used in biogas/fertilizer industry and animal feed industry |

With an increase in the world population, the demand for meat products is raising which rapidly deteriorates the environment and resources. The by-product percentage found in the meat processing industry is shown in Table 2.

**Table 2.** By product percentage of the meat processing industry.

| | | By-Product Percentage | References |
|---|---|---|---|
| Abattoirs | Cattle | 40–60 | [16,21,24] |
| | Lamb | 45–65 | [21,25] |
| | Sheep | 25–35 | [26,27] |
| | Goat | 20.5–30 | [28,29] |
| | Beef | 30–42 | [24] |
| | Buffaloes | 65–75 | [11,16] |
| | Pork | 47–61 | [21,30] |
| Poultry | Chicken | 10–45 | [16,31] |
| Fish processing industry | Fish | 30–80 | [32,33] |

## 3. Current Disposal Techniques of Slaughterhouse By-Products

The disposals of animal by-products are of critical concern not only for developing countries but also globally, as almost more than half of the by-products are discarded and transported to the landfill site. This incurs not only levy for the government and its transportation costs but harms the development of slaughterhouse processing industry

establishments [34,35]. Slaughterhouse by-products that are generated from the abattoirs processing industry are either disposed to landfill, incinerate, or buried. The solid by-products are classified as solid or liquid wastes [34,36].

The ranking of the management techniques for any type of by-products is shown in Figure 1. The by-products are any types of secondary products that are generated from the slaughterhouse. Mostly according to its desirability, it is recommended for people to reuse, recycle, and recover or reduce [34]. The hierarchy explores the maximum benefits of the meat processing industry [35,37]. Among those different waste disposal techniques, landfills are not encouraged due to their high environmental pollution, especially in the case of slaughterhouse by-products [35,38].

The yield percentage of the by-products varies depending on the topographic area (breed) and ages of the animals, hence those by-products from cattle and buffaloes vary from 65–75% of the live load of the animal weight and the other animals account from 50–60% of the live weight and in poultry can reached to 30–40% of its live weight [39].

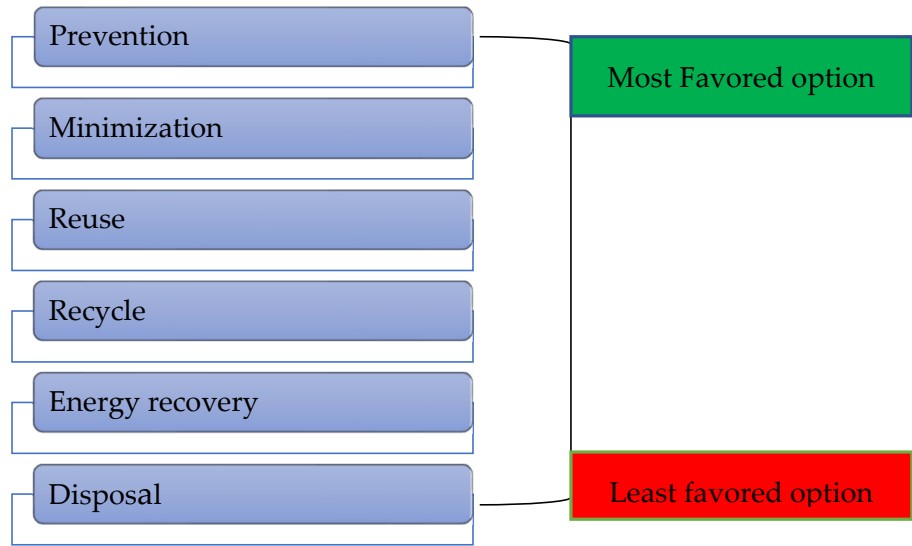

**Figure 1.** Slaughterhouse waste management hierarchy [40].

In the meat processing industry, the non-edible by-product contents of the slaughterhouse are most likely depending on religion, culture, health issues. Some people are not allowed to eat every edible part of the animals due to his/her medical issue [34,41,42]. All of the by-products (skin, hides, tails, horns, fetus, ears, trimming, hooves, blood, bone, gland nails, and bristles) which are not directly consumed by the customer are called non-edible by-products [42,43].

*3.1. Burial and Landfilling*

The landfilling technique is the most common type of disposal method and requires critical control to keep the odor, to minimize the environmental problem, groundwater contamination, and health issue [40,44]. Throwing off the by-product to the landfill site is a common practice in developing countries [45]. However, nowadays the use of landfills is quite less due to the population growth (lack of space availability) and the occurrence of landfill gases and methane causes contamination problems, groundwater pollution, soil contamination, and air pollution as well. It is strongly believed that the landfill technique has to be eliminated and changed to the production of high value-added products [46]. Dead animals and other rich sources of slaughterhouse biomaterials are buried in filled pits. Pits burial requires a confined environment (especially a soil environment) for sucking the non-carcass liquids as well as for preventing other liquids and fluids to speed up the anaerobic contamination at low humidity. Pits provides a confined soil environment

for absorbing carcass fluids and preventing heat loss, thus speeding up the anaerobic degradation process at the low moisture content [46].

### 3.2. Incineration

Incineration is one of the major types of thermal destruction of by-products by using heat to decompose the by-product materials in the existence of oxygen and most of this type is employed for the solid waste management system [34,47]. The merit of this system minimizes the space utilization of land by reducing the volume of solid waste by 90–93% [48]. For developed countries, incineration is the best option to minimize the space required to landfill [49,50]. Hence, incineration is subjected to thermal degradation of solid waste and converted to heat, gas, ash and steam. Burning of the solid waste material by temperature up to 850 °C and above would be performed for 20 min [5,34]. In a slaughterhouse, solid waste by-products are incinerated in a controlled manner. They may not cause expenses in the transportation of ash. Instead, they are used as a soil improvement [50].

## 4. Utilization of Slaughterhouse By-Product: Present Scenario

Abattoirs generate a large amount of solid as well as liquid waste and there is little experience on the utilization of this waste [51]. Most of the slaughterhouse processing industries have taken landfill of the by-products as a first option [52]. However, some of them are trying to recycle and reuse (for fertilizer) animal feedstock (as a glue agent for the wood industry, for pharmaceutical application, and decorative purposes) [10]. Meat plays a sustainable role in feeding of humans worldwide and one of the indications of meat consumption globally is the rate at which a given country is going to become rich. Indeed, more the 80 billion animals are slaughtered international for meat annually [53–55].

### 4.1. Slaughterhouse By-Product for Fertilizer

Slaughterhouse by-products are used as the production of fertilizer for crops such as soybean and could have positive merit on minimization of contamination and pollution problems (since the by-products are rich in nitrogen content that will help as a fertilizer) [44]. Composting is the best option for organic waste utilization instead of using landfilling and incineration techniques. Animal source materials such as manure and residues from the abattoirs by-product and most of the time manures are derived from mammalians and milk-producing animals when the animal is slaughtered only 40% up to 60% of the live animals are converted to usable product (market product). the remaining 40% to 60% taken as by-products [56]. Those by-products taken from slaughterhouses are mostly inedible such as feathers, bone. Hides/skins, hoofs, horns and blood are the main source fertilizers such as blood meal, fish meal and feather meal [57]. Chicken manure with sawdust is the best organic fertilizer that is proposed to use for conditioning of soil than synthetic fertilizer. One by-product of the meat processing industry has a high organic fertilizer and high protein animal feed, N = 13.25%, $p$ = 1.0% and K = 0.6%, and this is a result of producing the only high source nitrogen instead of that of synthetic fertilizer (even if the fertilizers are compatible with the crops and biodegradable as well) [44]. The way fertilizers are produced from slaughterhouse by-products is shown in Figure 2.

Fertilizers productions from the meat processing industries are one option of utilizing the by-products to minimize environmental pollution and it is an economical method [58]. Due to having more nitrogen content, it is preferable to use for composting [59].

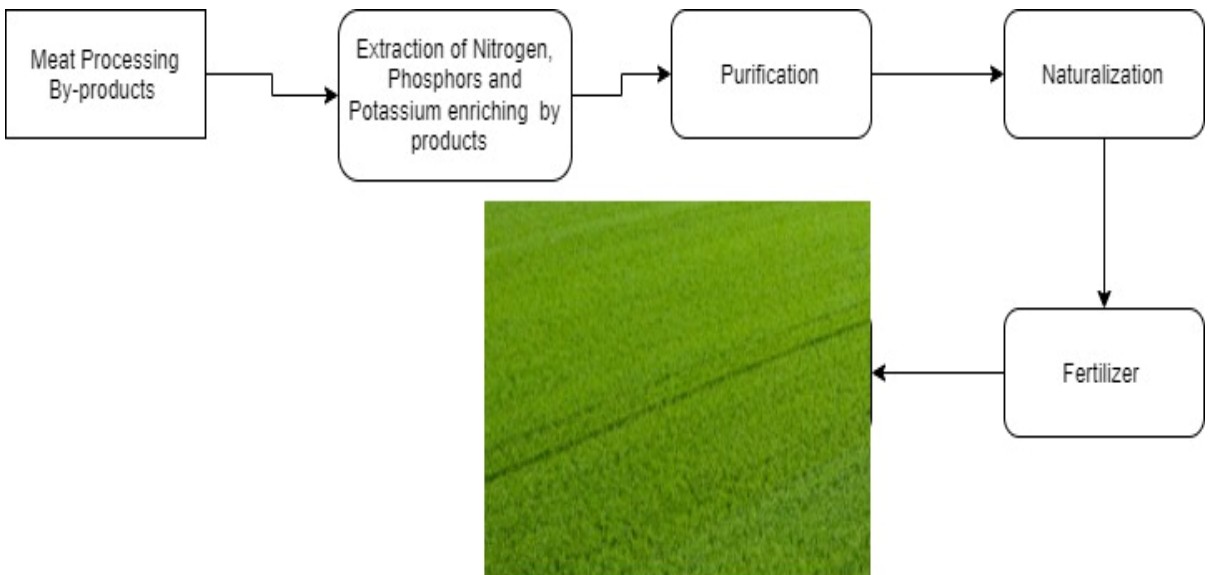

**Figure 2.** Process scheme of fertilizer production from the meat processing industry [44].

*4.2. Slaughterhouse By-Product for Animal Feed*

Most of the slaughter by-products are suitable for aforesaid application due to having mineral matter [16,60]. However, the best option is to administer its conversion into blood meal, bone meal, and fish meal. Those meals are utilized as food for chicken and fish [60]. The by-products have to be broken down into small particles for easy digestion by animals. Organic residues are preferably used for the production of animal feedstock [11]. Animal feedstock produced from the meat processing industry plays a great role in contributing to ensuring abundant organic sources, and affordable animal protein [61]. Biorefineries currently focus on the utilization of by-products (blood, bone) to produce animal feed by catering to the poultry and cattle segment as shown in Figure 3.

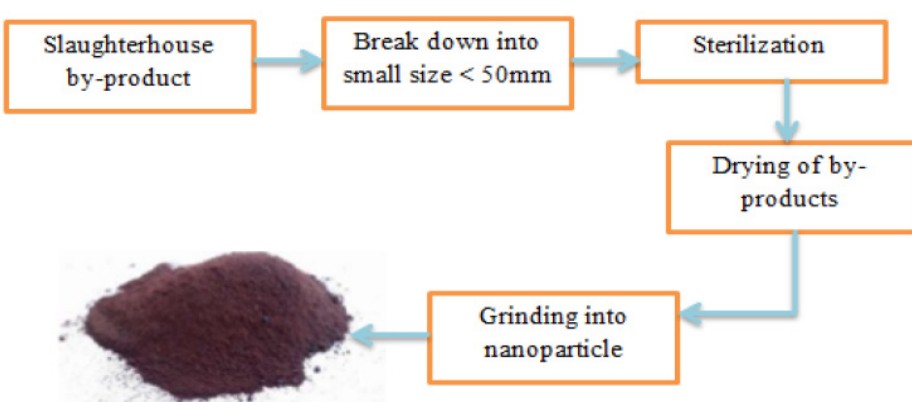

**Figure 3.** Animal feed production from the meat processing by-product industry [24].

*4.3. Slaughterhouse By-Product for Glue Application*

Globally, meat processing industries generate a large volume of by-products [60]. However, these by-products are highly utilized for industrial applications such as filler and glue [62,63]. The potential use of intestine, bone and skin, and fat proteins are converted into bioplastic and glue/adhesive agents for wood filler and bonding application [64]. The adhesives are produced by heating the listed by-products for 30 min with 60 to 90 °C in water for a PH of 5 to 9 and the production of glue from meat processing industries. The by-products are washed out to remove the dirt particles from it and soaked to make them soft enough then the stock is heated by boiling in open tanks under controlled pressure [63].

### 4.4. Slaughterhouse By-Product for Accessories

Materials that are disposed of to that landfills can be the starting point for producing any type of fashion accessories [65]. Nowadays the by-product amounts are converted to high value-added products chiefly in a fashionable manner in line with the demand of customer and economic level [66,67]. Fashion accessories are produced from any trimming wastes. For example, intestines are often utilized for the application of a musical instrument [68]. The horn is the main by-product of meat processing industries but in developing countries by combining the skin/hide and the horn they produce a fashionable chair with a cheaper cost. From the hoofs, foot oils are produced for shoe shining, neat and maintain the quality as well [68]. Blood waste is 3.5% of the live weight of animals and is used in leather finishing process [65]. Fashion products such as seats, lyres, and jewelry are produced from the trimming waste of the meat processing industry.

## 5. Physiochemical Properties

Knowing the physical and chemical characteristics of the meat processing by-products is an essential precondition for the selection of disposal technique, technology process used to utilize it and understanding ways of easily handling of by-products as well [69]. Understanding the chemical compositions of the by-products is the best way to select the produced high-value product by biorefinery approach [70].

### 5.1. Physical Properties

There are different parts of slaughtering such as edible meat, edible by-product and waste. Edible by-products can be kidney, heart, tongue, liver, stomach and intestine [11,71]. Solid wastes are the testicles, penis, tail, tallow, lung, skin/hide, tail, bone (shank) and head [71]. Liquid wastes are blood, bile, and feces [69]. The potential waste category is solid waste and the potential solid waste type is tallow [69]. Almost all of the solid wastes and by-products are transported to the landfill site. The blood and the stomach content are discarded into the surroundings and the other wastes are given to the customer with their carcasses application [11]. The discarded waste creates unpleasant odor in the environment and the people have protested [69,72]. There is no recycling method in the slaughtering house to eliminate or minimize the problem [73]. The percent yield used in Ethiopia of the by-products found in the meat processing industry is shown in Table 3.

**Table 3.** The yield of animal meat processing industry by-products [55].

| By Product/Meat | Yield Percentage | References |
|---|---|---|
| Boneless meat | 28.5–32 | [22,55,56] |
| By product/waste percent | 16 | [22,55] |
| Bone, head, and feat | 23–25 | [55,57] |
| Hide | 7–8 | [55,74,75] |
| Lunges and blood | 5–5.5 | [22,55] |
| Liver and stomach | 2–2.5 | [55,57] |
| Head meat and brain | 0.2–0.4 | [22,55] |
| Heart, tongue, spleen, kidney, and far | 2.5–3.3 | [22,55] |
| Casings, Urine, bile, dung, and body fluid | 5.1–5.5 | [55,56] |
| Other red, white, and dark offal | 4 | [55] |

### 5.2. Compositions

The chemical compositions of the slaughterhouse by-products are amino acids and nitrogen. This is an indication of the production of animal feed and fertilizers and there is an input for the production of high-value products such as biogas, bioenergy and medical application [11] Among the slaughterhouse by-products feathers and blood are the main

sources of protein and legs and heads in addition to protein there is a high amount of fates [74,76]. In general, the animal live weights are characterized by edible non-edible and discarded by-products amongst them more than 50% of the by-products are not consumed by humans due to different reasons [77]. Characterization of the slaughterhouse by-product in terms of protein, mineral matter, water and fats are critical issue to know the impact of each by-product type on environmental, health and economic issues as the same time knowing the amino acid compositions and contents are a good indication to produce the high-value by-products by using biorefinery approach [78,79]. Commonly, the meat processing industries are characterized as protein, mineral, fat and water and its percentage is found in Table 4.

**Table 4.** Typical product quantities after rendering 1000 kg of various slaughterhouses by-products [80].

| Raw Material | Quantity (Kg) | Proteins (Kg) | Mineral Matter (Kg) | Fat (Kg) | Water (Kg) |
|---|---|---|---|---|---|
| Animal carcasses | 1000 | 278 | 72 | 236 | 696 |
| Slaughterhouse waste | 1000 | 180 | 40 | 274 | 748 |
| Bones, blood, and bristles | 1000 | 241 | 127 | 47 | 573 |
| Poultry waste | 1000 | 124 | 32 | 181 | 663 |
| Fishery | 1000 | 280 | 7 | 23 | 690 |

## 6. Utilization of Slaughterhouse By-Product: Prospects

In the meat processing industry, every by-product should be utilized into high-value biomaterial if there is proper management of by-products characterization. Literature shows that the by-products percentage of buffaloes and castles ranges from 65–75% of a live load as compared with 50 to 60% of the live load of other animals such as sheep, goats, and pigs. However, the by-products yield from the poultry ranges from 30 to 40% of the live weight depends on the breed type and topographic area. In the abattoirs especially in Ethiopia, the majority of the wastes are produced during the slaughtering process. The slaughterhouse by-products are naturally protein-rich by-products which drive us to do on the characterization of it and utilization into high-value products as well.

Traditional slaughtering process found in Ethiopia are currently going to disappearing due to their environmental and health concern and the low price of the products themselves. Food and agricultural organization in the United States says that in 2030 the predicted meat production found in developing countries has to be 250 million tonnes among them almost 125–165 million tonnes are taken as by-products for the reason these by-products give us a potential area for the production of high-value by-products by biorefinery approach and the utilization of the by-products have importance to create better returns to the producer, reduce environmental pollution, employment generation, Increased soil fertility and conservation of resources [81]. The by-products from the meat processing industries are currently utilized as compositing, rendering and glue applications [82]. Some of the examples of how to use by-products in high-value products or biomaterials are discussed.

### 6.1. Biogas Production

Biogas is a renewable energy option produced through anaerobic digestion of wastes such as livestock excreta and slaughterhouse wastes. Biogas production is treated as one of the leading processes to combat climate change as well as a waste management strategy, especially for a developing country such as Ethiopia. It is also vital in meeting future demand for energy utilizing of indigenous sources [83,84]. Biogases are produced from by-products microbiologically through anaerobic fermentation. Livestock and poultry wastes are used for the production of energy in biomass since the wastes are rich in organic matter and most of the time fresh waste is more recommended and most suitable for its production [81,82]. Some of the by-products used for the production of biogases are the intestine; stomach and blood are the favored source of production [85,86]. Biogas

is produced mainly from organic residues and it consists of 45–85 vol% methane ($CH_4$) and 25–50 vol% carbon dioxide ($CO_2$) [87]. The biogas which is produced from the meat processing industry is a well-known method not only for the production of energy but also minimizes the waste amount which is transported to landfills [88].

Biomass in the world is the only source of renewable energy which brings electricity; provide heating, liquid and gases, cooling and fuel in form of solid. The meat processing industry globally supplies almost 11.5% of the principal vitality and about 79.9% of the international energy consumption. In the year 2012 around 194.8 million tons of the principal energy was spent. However, the biogases are produced by anaerobic fermentation of by-products. The biogas production capacity will reach 25,000 MW by 2026 [89]. Figure 4 below shows how biogas is produced from animal organic waste [90].

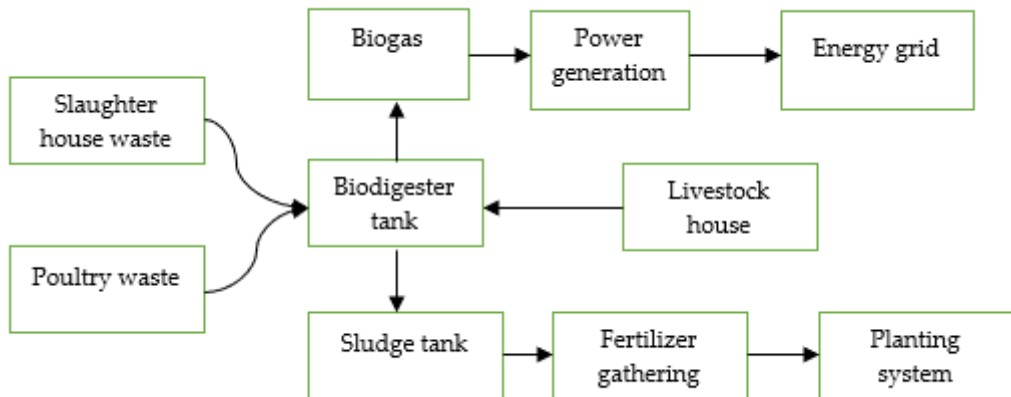

**Figure 4.** Potential for biogas production from slaughterhouse residues [86].

The ruminants from the meat processing industry offer a dominant resource worldwide and could be used for the production of biogas using fertilizer [87]. For the production of biogas from the slaughter by-product, there is 50%, 30%, and 99% availability of livestock, poultry, and sheep, respectively [75]. The amount of biogas yielded from the meat processing industry is found in Table 5.

**Table 5.** The amount of biogas produced from different substrates.

| Substrates | Percentage of By-Products | Biogas Yield (m³/Kg) |
|---|---|---|
| By product from cattle (fresh) | 25–30 | 0.6–0.8 |
| By-products from sheep (Fresh) | 18–25 | 0.3–0.4 |
| Byproducts from poultry | 10–29 | 0.3–0.8 |
| Blood Liquid | 18 | 0.3–0.6 |
| Rumen Content | 12–16 | 0.3–0.6 |

*6.2. Keratin*

In meat processing, by-product keratins are mostly found in the hair, wool, nail, hoofs, horns, bones and feathers [91,92]. Amongst the biopolymers found in the meat processing industry by-product keratin is the greatest plentiful source for the production of profitable and sustainable innovative materials as well are found in Figure 5.

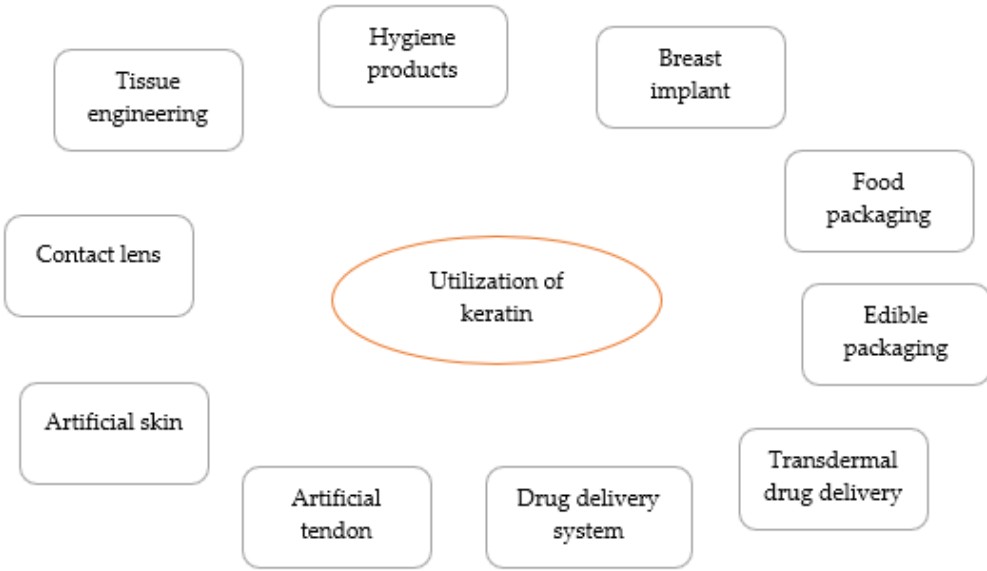

**Figure 5.** Possible utilization of keratin [92].

6.2.1. Edible Packaging

Among the biomaterials found in the meat processing industry keratin is the most abundant one although many reviewers deal on the utilization of keratin for medical, fertilizer and cosmetic application it has been seen in the utilization of keratin for the production of edible packaging which is the most favored food [93]. Edible packages are produced from composites, lipids, resins, and hydrocolloids [94]. The reason for increasing interest and research work on edible packaging is due to stable food and the convenient impact of biodegradable waste of packaging on the environment [95] and keratin application is found at Figure 6.

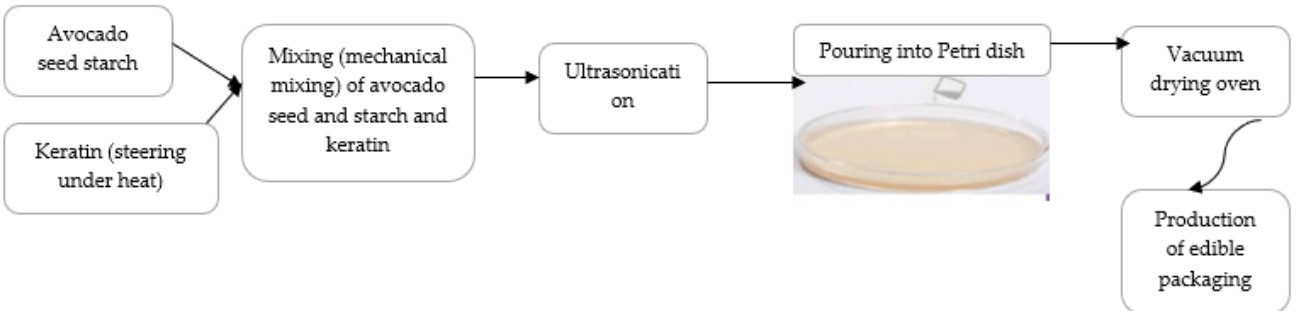

**Figure 6.** Utilization of keratin for production of edible packaging [95].

6.2.2. Tissue Engineering

Many reviews have been carried out on utilization and characterization of keratin biomaterials due to their biocompatibility and biodegradability and positive impact on the body of humans until now few of those researches' biomaterial developments are applied and used in tissue engineering [61]. Scaffolds produced from keratin of animal hair shows that simplifying the peripheral nerve regeneration and encourage neurovascular retrieval and act efficiently in the hemostatic agent [85]. The capability of keratin-built biomaterials interpreted into the people medical setting is highly focused on further research work to elaborate on the mechanisms regarding hemostasis and nerve recovery [16] and utilization of keratin for tissue engineering is found in Figure 7.

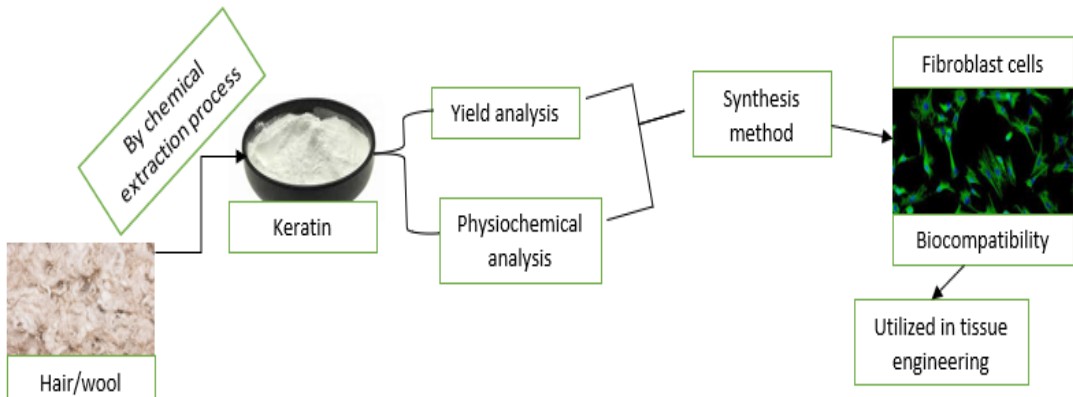

**Figure 7.** Utilization of keratin for tissue engineering [61].

### 6.2.3. Drug Delivery

For numerous drugs, tenders-controlled drug delivery has been a prerequisite to have therapeutic efficiency and to avoid its adverse effects [96,97]. Keratins are the best biomaterial for drug delivery of medicals for humans due to their compatibility, safe for the human body and demanded materials for health [98,99]. The utilization of drugs by incorporating nanomaterials is also another option to deliver drug, and due to its biocompatibility, easy to process, non-toxicity and biodegradability, keratin is very demanded biomaterial since any biomaterial for the drug delivery system must fulfil several requirements [98] and the details are found at Figure 8.

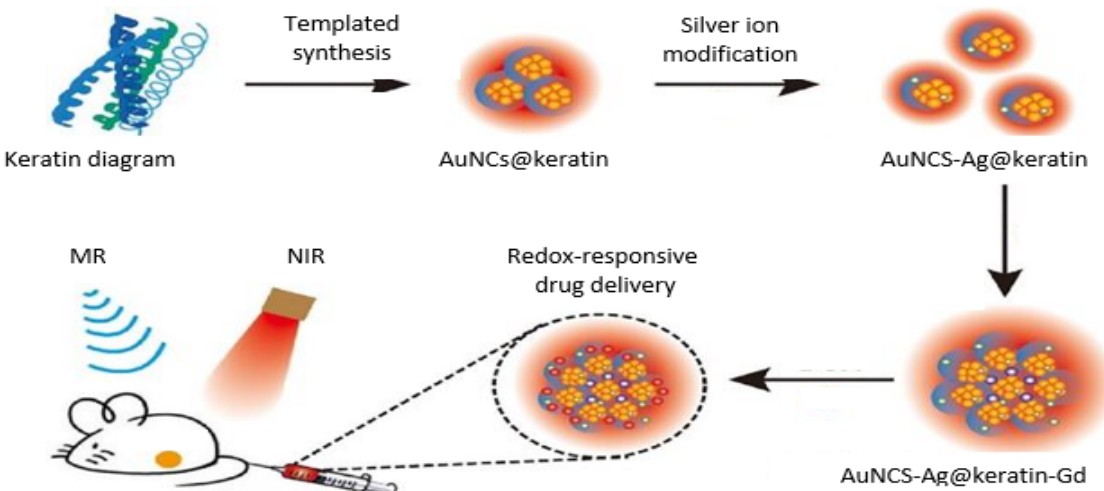

**Figure 8.** Keratin-templated formation of the gold nanoclusters, with silver and gadolinium, for drug delivery system [74].

### 6.2.4. For Textile Application

Among slaughterhouse by-products, the hoofs, horn, and bones are the main sources of keratin and gelatine. However, those by-products by themselves could not be reacted with textile materials especially with cellulose so the keratin should be dissolved and transformed to keratin reactive hydrolysate [65]. Then, the hydrolyzed keratin is applied for the conventional dyeing and finishing process as well [65,100].

The keratin content of bone, hoof and horn are around 93.3% and a Nemours's amounts of keratins are discarded to landfill without any utilization from hair, hoof, hair and bones [101] and it can be classified as soft (1% sulphur) and hard (5% sulphur) based on the sulphur content. Keratins are broadly classified as hard (5% sulphur) and soft (1% sulphur) keratins hence due to the tight packing of the protein chain in the form of B-sheet and

ά-helix into coiled polypeptide chains are cross-linked to disulphide bonds the keratins are chemical unreactive with textiles mostly in feathers B-keratins are found whereas ά-helix are found in horns, hoofs and hair [102,103]. Salt is the main exhaustive agent in the dying process of textile materials with reactive dye but it has a defined problem since it increases the effluent load in terms of total dissolved solids and it limits the recycling of the effluents released from the dyeing industry [100]. In order to producing blood-stained fabric, the feathers are first fermented at optimal condition for getting keratinase and amino acid through fermentation broth and then the keratinous materials are mixed with detergents for achieving distain process of the fabric.

### 6.2.5. Medical Application

The utilization of keratin for biomedical application and due to its plentiful, bioactive and representative source of proteins know it is taken as clinically relevant biomaterials [91,104]. Keratin is cysteine reach protein and highly plentiful and found mostly in horns, wool, feathers and nails due to its special nature in case of absorbing metals and some compounds was used highly for purification purposes [91]. A paper shows that keratins are used for medical applications especially for controlled drug and tissue engineering due to the ability to capture hydrophilic and hydrophobic drugs [105,106]. Furthermore, animal organs can also have medical applications [16,107]

### 6.2.6. Fire Extinguishers

A fire extinguisher requires a chemical which is nitrogen water and air (foam extinguisher) [108]. Fire extinguishers are typically used in food industries, textile industries and medical industries and allow the workers, to emancipation the extinguisher, renew it, and return to the fire within a time [109]. Nowadays, some unconventional fire extinguisher materials are used. The meat processing industries horn and hoof is a rich source of foaming used in fire extinguishers especially in airplanes [108]. The keratins found in horns and hooves are created bonds with the foam froths combined into a robust blanket and avoid it from contravention on impact with the fire (which is why keratins are preferable for fire extinguisher applications) [110].

### 6.3. Collagen

The application of collagen is numerous for medical application and found from especially bovine and [111,112]. Hydrolyzed gelatin is a fibrous protein found in the tissue of animals and extracellular locations. in mammalians, 25–30% of the proteins are found in collagen and the main sources of collagen among the meat processing by-products are corneas, bones, blood vessels, cartilage, and the dentin of teeth, skin and hide [112]. Collages are found from many sources mainly from marine organisms, nails of animals and humans, land organisms, and has to be produced through hydrolyzed process.

The structure of collagen is composed of three polypeptide chains in the form of a triple helix among them the two are identical chains and the third are somewhat its chemical composition is different. collagen is not only found in the skin/hide of animals but when the age of humans goes up the collagen content going to decrease by 15–25% and it is the main structural protein in the various connective tissues in the skin and bone. Collagen is the most common type found in mammals making up from 25–35% of the whole-body protein content [112].

### 6.3.1. Skin Revitalization

As collagen is found in the skin and bone of meat processing industries it is preferable to use skin revitalization due to its biodegradable and compatible with the human body and 80% of the collagen found in the human skin is protein. to keep the moisture of the skin by providing the elasticity, protection and structure as well. the collagen content decreases over time naturally and its reduction of collagen the case for the skin converts to thinner, the resistance is minimized, drier, shirk of skin.

### 6.3.2. Medical Application

Collagens are mainly used in tissue engineering and for tissue regeneration scaffold is required [113]. Collagen scaffold used for different purposes mainly to visualize cells in the nervous system which helps for bone reconstruction. Scaffolds could be used for implantation during grafting failure and when the faults raise at the maximum level [113]. Hydrolyzed gelatins are excellent biocompatibility due to pathetic antigenic and easy to biodegradability among the collagen source are make bovine is more demanded medical application.

Collagens are prepared in the form of sponges for wounds or burns in the form of film and powder for surgical sutures [114]. Hydrolyzed gelatin or collagen has numerous applications in the field of medicine such as in heart valves (cardiology), skin replacement (skin tissue engineering and artificial skin dermis), for wound repair and dressing, ligament and bone repair, and for cartilage reconstruction [115]. There are five different types of collagens found in different body parts of the animal's collagen. Type one is in the bone, ligaments, tendons and skin. Collagen type two is found in the eyes and cartilage. Collagen type III is found on livers, lunges and arteries. Collagen type four is also found in the kidneys. Collagen type V is present on the surface of cells, hair and placenta [111]. Due to its biodegradability, biocompatibility, non-toxicity, inherent, non-irritating and cheapness, gelatin or hydrolyzed collagen are used in the manufacturing of drug delivery as well the carboxylic and the amine group in gelatine makes a great modification of structure [111]. Collagen used for dental and wound care application.

### 6.3.3. Pharmaceutical Industries

Collagens' application in pharmaceutical as a biomedical sector is a result of its biodegradability, biocompatibility and weak antigenic properties. However, in pharmaceutical companies as microparticles and drug delivery [114]. Mainly, the hydrolyzed gelatin or collagens are found on different parts of the animals but are typically found in cattle skin and pig skins and among the slaughterhouse and poultry by-products collagens are found in fish in more and there is no religious restriction on the utilization of fish collagen as a pharmacy [116]. Thus, the pharmacies produced from the collagens are tablets, shields, sheets, scaffolds, films, and microspheres [117]. Hydrolyzed collagen and gelatin collagen and it has so many advantages such as Advances health of skin, hair, nails, reducing joint pain and degeneration, healing the gut, boosts metabolism and keeping calm [114].

Pharmaceutical application from the meat processing industry is a new technology [118]. By-product materials have many valuable nutrients. Some of them are used for medical applications due to their having amino acids, hormones, minerals, vitamins, fatty acids and cysteine [16]. The by-products such as lung, kidney, brains, spleen, tripe and intestine have more moisture content than edible meat such as blood among this kidney and lung have more carbohydrate contents than meat [119]. Due to these kidneys and lung are converted into high-value-added materials such as pharmaceuticals [120]. Most of the blood components such as fibrinogen, fibrinolysin, serotonin, kalikreninsa, immunoglobulins and plasminogen are utilized for chemical or medical uses [16].

### 6.4. Cosmetics Application

In a global market currently, the skincare products produced from the fish are preferable due to moisture-retaining functions [121]. Amino acid surfactants are biosurfactants used in many personal care product formulations and preferred from surfactants that produce synthetic materials. Hydrolyzed surfactants found from fish are used for many uses such as hair softening, rinses; shampoo hair treatment agents at the same time reduce the harshness and breakage of hairs by penetrating throw-out the cuticle into the hair [122,123].

The main raw materials used for the production of cosmetics are fats, oils, wax and ester oils. Some surface-active agents and surfactants are used for emulsifier purposes. However, the meat processing industry by-products due to high-fat content are valuable materials for cosmetics production as well. Cosmetics produced from the meat processing

by-products are mostly preferred to use due to their compatibility with the body of humans. The processes for the production of cosmetics are as follow in Figure 9.

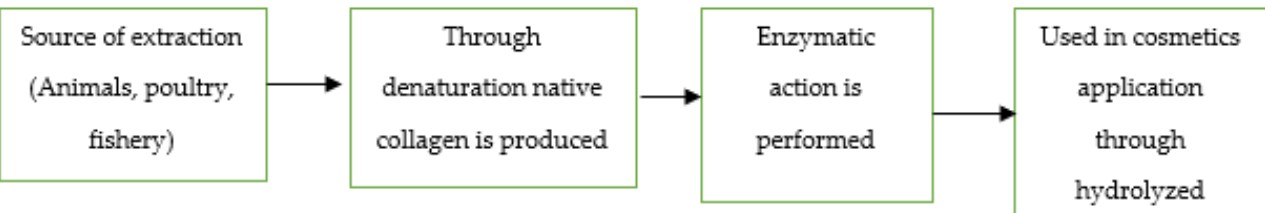

**Figure 9.** Production scheme of cosmetics from meat processing by-products [122].

*6.5. Food Industry*

Globally, the meat processing industry's by-products are utilized for the production of edible by-products for human use since they have high protein levels and mineral iron content. The slaughterhouse blood having of 80.9% water, 0.62% mineral, 0.23% lipids and 17.3% protein of the live weight of the bovine animals. In developed countries, the utilization of blood for blood sausages, blood pudding, biscuit, blood curd and breed have a long history [124]. Blood by-products are not only used for the production of human consumption products but also non-food items such as fertilizer, binders for textile application and feedstuffs for animals and the production of emulsifier, stabilizer and color additives in textile dyeing industry and the blood amount account (2.4–8)% of the live weight of the animals [39].

Midst of the animal by-products hides and bones are the main sources of gelatins and used for the production of food ingredients and dessert food too for the reason that of the elasticity, consistency and immovability used widely and besides, in the food industry, it is used also for medical pharmaceutical and photographic industry [78]. Gelatin is produced by hydrolysis of the collagen investigated from cartilages, bones, tendons, skins and bones of animals, and most gelatins are produced from pigs. However, due to cultural issue, the Islamic religion, and various health issue, the pigs' by-product is not produced from it [8,125]. Its chemical structure makes it non-digestible and films made from it are more transparent and flexible (which makes them useful for food processing and paper production). The process for gelatin extracted from the meat processing industry is found in Figure 10.

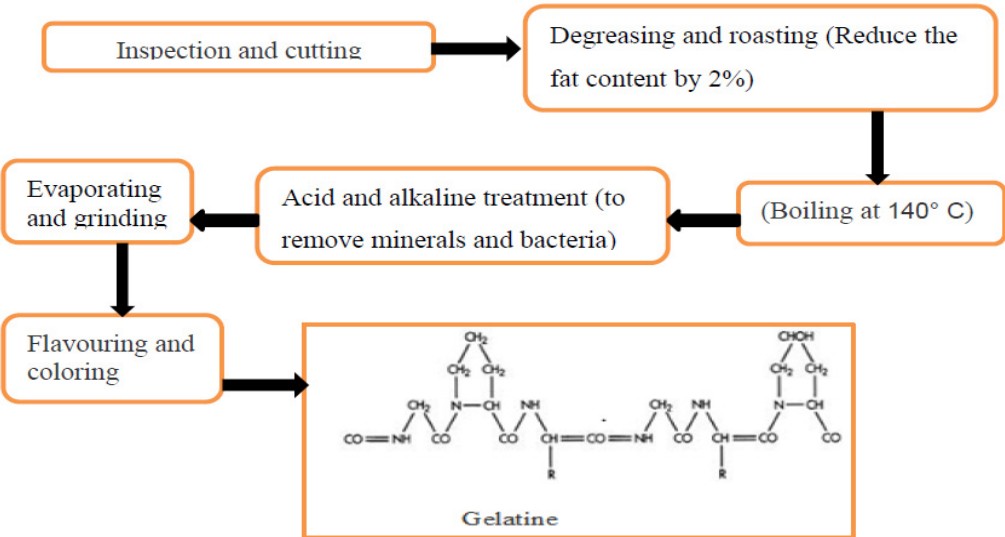

**Figure 10.** Production scheme of gelatine from slaughterhouse by-product [125].

### 6.6. Surgical Sutures

Surgical suture is medical items used to grasp the body tissue organized after an injury by different issue [126,127]. Surgical suture of humans requires a needle and thread which is compatible with the body of the humans or animals. Intestines are the main compatible material for surgical sutures from the meat processing industry by-products due to its flexibility strength, rapidity of insertion and absorbability [128]. Numerous investigational studies can be found in the literature on the utilization of animal intestines for surgical sutures in literature, in terms of healing, post-operative narrowing of the intestine lumen and how to form adhesion [129]. To minimize the risk of humans made by thread nowadays the best option is the utilization of the intestine in terms of cost and compatibility to their body and how to utilize the intestine for surgical application.

## 7. Conclusions

The meat processing industries produce a large volume of solid and liquid waste, and from 50–65% of the live weight of the animals is not consumed by humans due cultural issue, religious issue and health problems and their disposal are either incineration or landfill which facilitates the environmental impact and contamination due to wastewater, spent air/waste gases, noise, animal waste, waste heat and residues in the end product. Those by-products are incredible resources of protein, fat, keratin, collagen, gelatine and mineral matter that could be used for the production of high-value biomaterial, biochemical and by-products. The utilization of by-products in cosmetics, biofuels, biodiesel, tissue engineering, textile and composite application not only minimize the waste disposal percentage but also there will be a great reduction of soil degradation, water contamination and depletion of space utilization as well. A wide focus on research and development is a necessity to improve their utilization of by-products for the right application as explored in the review.

**Author Contributions:** Conceptualization, D.Y.L. and T.T.; methodology D.Y.L.; software, D.Y.L. and T.T.; validation, D.Y.L., T.T., M.A., N.M.H. and W.M.; formal analysis, D.Y.L.; investigation, D.Y.L., T.T., M.A. and N.M.H.; resources, D.Y.L., E.F., A.H., A.A. and G.G.G.; data curation, D.Y.L., T.T.; writing—original draft preparation, D.Y.L.; writing—review and editing, D.Y.L., T.T., M.G. and F.K.; visualization, D.Y.L.; supervision, T.T.; project administration, D.Y.L. and T.T.; funding acquisition, D.Y.L. and T.T. All authors have read and agreed to the published version of the manuscript.

**Funding:** The Higher Education and TVET program Ethiopia-Phase 3, PE479-Higher Education, KFW project (No. 51235) and BMZ (No. 201166305) for the financial support of this research and National Foreign Expert Program–China (Grant No. DL2021024001).

**Institutional Review Board Statement:** Not applicable.

**Informed Consent Statement:** Not applicable.

**Data Availability Statement:** Not applicable.

**Acknowledgments:** The authors would like to acknowledge the Higher Education and TVET program Ethiopia-Phase 3, PE479-Higher Education, KFW project (No. 51235) and BMZ (No. 201166305) for the financial support of this research and National Foreign Expert Program–China (Grant No. DL2021024001).

**Conflicts of Interest:** The authors declare no competing interests.

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
