# Peer review of "A Comprehensive Review on Utilization of Slaughterhouse By-Product: Current Status and Prospect"

_sustainability, doi:10.3390/su14116469_

Round 1

Reviewer 1 Report

The subject of the study is very interesting. Unfortunately, the quality of the manuscript is very low. What is the main aim of this review?

There are some major changes I am suggesting in detailed comments below.

Detailed comments:

All figures are of poor quality. The information of the figures seems interesting but the authors do not refer to them in the text, so the question is whether they are needed?

Some figures do not have a specific source of literature. Does it mean that the authors made them themselves? (Figures 1, 3 (very bad quality, unacceptable !, 5, 10, 13, 14).

The format of the tables needs to be improved.

Please format the references in a uniform way and according to the requirements of the journal

Table 4 There is in the second column " percentage" - please add the percentage of what (CH4%?).

Author Response

Response to Reviewer 1 Comments

Point 1: The subject of the study is very interesting. Unfortunately, the quality of the manuscript is very low. What is the main aim of this review?

Response 1: The main aim of the review paper is to show the possible utilization routes of slaughterhouse by-products, and in the Ethiopian case the current disposal methodologies of these by-products are problematic, contributing to environmental contamination, soil degradation, air pollution, and possible health problems Nevertheless, those by-products are rich in collagen, keratin, and minerals being thus promising sources of high-value materials such as bioenergy, biochemical and other biomaterials that could be exploited in various industrial applications”.

Point 2: All figures are of poor quality. The information of the figures seems interesting but the authors do not refer to them in the text, so the question is whether they are needed?

Response 2: we have referred to all the figures in the manuscript as well so that the figures are too needed

Point 3: Some figures do not have a specific source of literature. Does it mean that the authors made them themselves? (Figures 1, 3 (very bad quality, unacceptable !, 5, 10, 13, 14).

Response 3: the quality of the figures are changed by using an online diagram maker system

Point 4: The format of the tables needs to be improved.

Response 4: the format of the tables are edited based on the journal guideline

Point 5: Please format the references in a uniform way and according to the requirements of the journal.

Response 5: we have checked and made a correction on the referencing of the manuscript

Point 6: Table 4 There is in the second column " percentage" - please add the percentage of what (CH4%?).

Response 6: the percentage of by-products and from that amount of by-product only a little amount can be utilized for the production of biogas

Reviewer 2 Report

The topic is interesting and the structure in general adequate. The language, as well as the formatting (figures and tables), need polishing though! Specific comments below

Line 3: ‘Prospects’ instead of ‘prospect’

Line 14: ‘reaching’ instead of ‘that is’

The abstract needs to be revised extensively. Suggestions, below

Lines 15-23: The live weight of the animals is distinguished in edible, inedible, and discardable by-products, with the discardable parts reaching of the initial live weigh for cattle, lamb, and pigs, respectively. A small percentage only of those of by-products nowadays are exploited for the production of high added values products such as animal feed, glue, fertilizers, etc., whereas the main management method is the direct disposed to landfills. As such, the current disposal methodologies of these by-products are problematic, contributing to environmental contamination, soil degradation, air pollution, and possible health problems Nevertheless, those by-products are rich in collagen, keratin, and minerals being thus promising sources of high-value materials such as bioenergy, biochemical and other biomaterials that could be exploited in various industrial applications.”

Instead of “The disposal techniques of these by-products have a great problem on environmental contamination, soil degradation, air pollution, and health problem. A little amount of by-products now a days are utilized for the production of animal feed, glue, fertilizer, and the remaining are disposed to landfill. While those by-products are rich sources of collagen, keratin, and mineral matter they are also the best sources of high-value materials such as bioenergy, biochemicals, biomaterials, biomedicine, and industrial applications. Moreover, more than 50% of  the slaughterhouse by-products are not directly consumed by people. The live weight of the animals is characterized as edible, inedible, and discarded by-products among them the by-product of cattle, lamb, and pigs represent 66%, 52%, and 80% respective

Lines 23-27: “In this paper, the possible utilization of slaughterhouse by-products for the production of different high added value materials is discussed. Into this context, the various processes presented provide solutions to more sustainable management of the slaughterhouse industry, contributing to the reduction of environmental degradation via soil and water pollution, the avoidance of space depletion due to landfills, and the development of a green economy.”

Instead of “In this paper, the potential resources and possible utilization of slaughterhouse by-products for value-added products are discussed. Accordingly, the conversions of by-products to high-value biomaterial reduce the environmental pollution, soil degradation, and water contamination besides depletion of space for landfill and create sustainable development and green economy.”

Line 31: Please add Great consideration has been given lately

Line 32: Please revise “instead of having landfilling, incineration, and burial to minimize” e.g. to “instead of applying landfilling, incineration, and burial, aiming to minimize…”

Lines 36-37: which problems? You haven’t referred to specific problems above

Lines 52-53: the values provided here are different from those given in the abstract! Please check

Line 58: the requirement of meat for consumption

Line 59: “increase” instead of speed up

Line 59: Please revise. The meaning is not clear

Lines 60-61: “The majority of studies on the possible valorization of slaughterhouse by-products deal with their conversion to animal feeds and fertilizers.” instead of “Most of the scholars deal with the utilization of slaughterhouse processing to animal feeding and fertilizer”

Lines 61-62: “The current work goes a step beyond, reviewing possible alternative ways for the utilization of by-products into various high-added-value biomaterials.” Instead of “however; the current work reviews the possible roots in the utilization of by-products in high value-added biomaterials.”

Line 65-68: “…their conversion into high-value products … industries [10,11] is proposed.” instead of  “…convert them into high-value products …industries [10,11].”

Line 80: reach nearly 150 million tonnes

Lines 91-92: Please revise

Lines 95-96: please omit

Line 106: the natural resources

Line 109: 3. Current Disposal Techniques of Slaughterhouse By-Products

Line 110: are of critical

Line 117: The ranking of the management techniques for any type of by-products…

Figure 2: Is of poor quality (looks flattened). Please revise

Figures in general: Please use the same font type and size in all figures so as to the format to be common!!

Line 218: disposed to landfills

Line 229: 4. Physiochemical Properties of slaughterhouse by-products

Lines 236-246: These are not physical properties. The types of by-products are described

Line 239-240: The meaning is not clear! Please revise

Table 4: Please revise the format

Lines 520-524: Too long sentence. Please revise

Author Response

Response to Reviewer 2 Comments

Point 1: Line 3: ‘Prospects’ instead of ‘prospect’?

Response 1: Prospect is changed by prospects ”.

Point 2: Line 14: ‘reaching’ instead of ‘that is?

Response 2: that is changed by reaching

Point 3: Instead of “The disposal techniques of these by-products have a great problem on environmental contamination, soil degradation, air pollution, and health problem. A little amount of by-products nowadays are utilized for the production of animal feed, glue, fertilizer, and the remaining are disposed to landfill. While those by-products are rich sources of collagen, keratin, and mineral matter they are also the best sources of high-value materials such as bioenergy, biochemicals, biomaterials, biomedicine, and industrial applications. Moreover, more than 50% of  the slaughterhouse by-products are not directly consumed by people. The live weight of the animals is characterized as edible, inedible, and discarded by-products among them the by-product of cattle, lamb, and pigs represent 66%, 52%, and 80% respective

Response 3: The live weight of the animals is distinguished in edible, inedible, and discardable by-products, with the discardable parts reaching of the initial live weigh for cattle, lamb, and pigs, respectively. A small percentage only of those of by-products nowadays are exploited for the production of high added values products such as animal feed, glue, fertilizers, etc., whereas the main management method is the direct disposed to landfills. As such, the current disposal methodologies of these by-products are problematic, contributing to environmental contamination, soil degradation, air pollution, and possible health problems Nevertheless, those by-products are rich in collagen, keratin, and minerals being thus promising sources of high-value materials such as bioenergy, biochemical and other biomaterials that could be exploited in various industrial applications.”

Point 4: Instead of “In this paper, the potential resources and possible utilization of slaughterhouse by-products for value-added products are discussed. Accordingly, the conversions of by-products to high-value biomaterial reduce the environmental pollution, soil degradation, and water contamination besides depletion of space for landfill and create sustainable development and green economy.”

Response 4 “In this paper, the possible utilization of slaughterhouse by-products for the production of different high added value materials is discussed. Into this context, the various processes presented provide solutions to more sustainable management of the slaughterhouse industry, contributing to the reduction of environmental degradation via soil and water pollution, the avoidance of space depletion due to landfills, and the development of a green economy.”

Point 5: Please add Great consideration has been given lately.

Response 5: lately been added on the document  

Point 6:  Please revise “instead of having landfilling, incineration, and burial to minimize” e.g. to “instead of applying landfilling, incineration, and burial, aiming to minimize

Response 6: instead of applying landfilling, incineration, and burial, aiming to minimize

Point 7: which problems? You haven’t referred to specific problems above

Response 7: to reduce environmental and health problems

Point 8: the values provided here are different from those given in the abstract! Please check

Response 8: We have checked and made a correction to the document  

Point 9:  Line 58: the requirement of meat for consumption

Response 9: The consumption of meat is increased as the population

Point 10: “increase” instead of speed up

Response 10: speed up is changed by an increase

Point 11: Line 59: Please revise. The meaning is not clear

Response 11: the sentence is revised

Point 12: instead of “Most of the scholars deal with the utilization of slaughterhouse processing to animal feeding and fertilizer”

Response 12: The majority of studies on the possible valorization of slaughterhouse by-products deal with their conversion to animal feeds and fertilizers.” 

Point 13: ” Instead of “however; the current work reviews the possible roots in the utilization of by-products in high value-added biomaterials.”

Response 13: The current work goes a step beyond, reviewing possible alternative ways for the utilization of by-products into various high-added-value biomaterials

Point 14: instead of  “…convert them into high-value products …industries [10,11].”

Response 14: their conversion into high-value products

Point 15: Line 80: reach nearly 150 million tonnes

Response 15: the by-product percentage found in the slaughterhouse is nearly 150 million tonnes worldwide

Point 16: Lines 91-92: Please revise

Response 16: the sentence is revised

Point 17: Line 106: the natural resources

Response 17: the topic is almost revised

Point 18: Line 109: Current Disposal Techniques of Slaughterhouse By-Products

Response 18: instead of slaughterhouse by-product disposal technique,  the current disposal techniques

Point 19: Line 110: are of critical

Response 19: edited

Point 20: Line 117: The ranking of the management techniques for any type of by-products…

Response 20:the sentence is paraphrased and edited

Point 21: Figure 2: Is of poor quality (looks flattened). Please revise

Response 21: the quality of the figure is checked and revised

Point 22: Figures in general: Please use the same font type and size in all figures so as to the format to be common!!

Response 22: We have used the same format and the font size is checked

Point 23: Line 218: disposed to landfills

Response 23: edited

Point 24: Line 229: 4. Physiochemical Properties of slaughterhouse by-products

Response 24: instead of Physiochemical Properties, Physiochemical Properties of slaughterhouse by-products

Point 25: Lines 236-246: These are not physical properties. The types of by-products are described

Response 25: the paragraph is revised

Point 26: Line 239-240: The meaning is not clear! Please revise

Response 26: revised

Point 27: Table 4: Please revise the format

Response 27: The format is edited

Point 28: Lines 520-524: Too long sentence. Please revise

Response 28: revised

Reviewer 3 Report

Please check the order of the citation. There are a lot of minor mistakes at work.
Page 2. There is no citation [12].
Page 2. Line 94. Add square brackets.
Page 4 You can replace citation numbers 40 with 39 so that they are consecutive.
Page 5 citation [38] is after citation [46], general mess.
Page 5. Line 159. Correct temperature unit.
Missing citation [50]?
Page 5. Line 170. Unnecessary comma.
Page 8. Line 256. Missing square brackets.
Page 8. No citation [76]?
Page 12. Line 348. Add square brackets.
No citation [97].
Are the drawings with numbers 5, 10, 13 entirely made by the authors of the publication?
The quality of the drawings should be improved.

Author Response

Response to Reviewer 3 Comments

Point 1: Please check the order of the citation. There are a lot of minor mistakes at work.

Response 1: we have checked and revised the citation of the manuscript
Point 2: There is no citation [12].

Response 2: reference number 12 is cited in the manuscript
Point 3: Line 94. Add square brackets.

Response 3: Square bracket is added
Point 4:  You can replace citation numbers 40 with 39 so that they are consecutive.

Response 4: the citation order is revised
Point 5:  Page 5 citation [38] is after citation [46], general mess.

Response 5:corrected
Point 6:  Page 5. Line 159. Correct temperature unit.

Response 6: the temperature unit is corrected
Point 7:  Missing citation [50]?

Response 7: revised

Point 8:  Page 5. Line 170. Unnecessary comma.

Response 8: corrected
Point 9:  Page 8. Line 256. Missing square brackets.

Response 9: The square bracket is added
Point 10:  Page 8. No citation [76]?

Response 10: reference number 76 is cited in the manuscript

Point 11:  Page 12. Line 348. Add square brackets.

Response 11: The square bracket is added
Point 12:  No citation [97].

Response 12:Cited
Point 13:  Are the drawings with numbers 5, 10, 13 entirely made by the authors of the publication?
The quality of the drawings should be improved.

Response 13: the quality of figures are improved by using an online diagram maker

Round 2

Reviewer 1 Report

Please format the references in a uniform way and according to the requirements of the journal.

Author Response

Point 1: Please format the references in a uniform way and according to the requirements of the journal.

Response:  The format of the reference is made uniform and edited the whole reference based on the journal standard. 

Reviewer 3 Report

My comments were taken into account.

Author Response

Point 1: My comments were taken into account.

Response: ok. 

This manuscript is a resubmission of an earlier submission. The following is a list of the peer review reports and author responses from that submission.

Round 1

Reviewer 1 Report

The topic is interesting since the slaughterhouse industry creates indeed several streams of by-products that can be vaporized via different processes, whereas the existing literature mainly deals with the exploitation of wastes and wastewaters. The approach and structure of the MS seem appropriate in general, but the MS seems to have been written in a rush. There are many repetitions of the same information, some information is irrelevant and the use of language is rather poor. The quality of the figures is in general poor, too.

Specific comments:

The introduction is rather confusing since the differentiation of by-products and actual wastes is not clear. Are the by-products considered as secondary products that can be exploited directly e.g. may be sold to other industrial sectors after separation? Are they discarded contributing to environmental degradation? Is the fate of those by-products different in various parts of the world? I believe these issues should be addressed clearly in the introduction. Moreover, the novelty statement should be emphasized. Which gaps of the existing literature do the study aspires to fill? What is its main aim?

Paragr. 1.1. Overview… I think that in this paragraph you should briefly mention the different sectors of the shlaughterhouse industry (meat, poultry) providing numerical and statistical data about the global distribution, handling of product,s and general types of by-products. Moreover here the different streams of by-products generated from the different types of slaughterhouses should be listed as well as the stage at which they are produced.

Lines 97-107: I believe the water demand is out of scope.

Line 114: please add references

Figure 2: I do not think that the scheme is needed. It can be omitted.

 An issue of great importance that is not emphasized in the study is the safety of using slaughterhouse by-products

Table 1: Please format the table. The fond size and type should be revised.

Figure 3: I think that “Dressed carcass”, should be either “dressed weigh” or “carcass weigh” and it is not only the meat. It also includes bones, cartilage etc.

Line 148: What do you mean by “…the waste are quantified…”

Lines 151-153: Out of scope. Wastewater is not by-product

Line 155: I think that you mean the opposite of what is written here. Landfill is the least desirable option

Line 170: …by products. Actually…

Line 175: are called, instead of we call them

Line 180-193: So landfilling is just throwing away, digging in pits or incineration?

Lines 184-185: repetition of info provided above

Lines 217-219: Repetitions again

Paragr. 3.1: The whole paragraph should be rewritten in clearer way. Which by-products can be used for the production of fertiliser? How about its final use? How is sanitation ensured? Manure is certainly not included in the slaughter by-products as presented in Figure 3. Figure 5 does not provide any important information. Please omit

Lines 242-245: Very generic. Please elucidate

Parag. 3.2. Very generic content.  Which by-products can be used for animal feeds and for which animals? Are there restrictions on their use? How about their sanitation? Fig. 6 is also generic but provides useful info. Please elucidate, providing more detailed information on the steps presented.

Line 248: The meaning is unclear and it sounds as contradiction to the previous paragraph. Please rewrite

Line 249-250: blood meal is mentioned twice

Line 251: Do you mean, have to be hydrolysed to be more easily digested…?

Par. 3.3. Very generic again. Which parts exactly are suitable for glue production and why? The proteins? What are the chemical compounds that are valorised and which are the reagents needed during processing (referring as “ingredients” in Fig. 7).

Figure 7 seems to be incomplete. How are the parts connected?. What does the And stand for. Which are the “ingredients”. What is “mechanical filtration”?

Par. 3.4. I think the title is misleading. Figure 8 has no meaning. Please omit. Line 279: Are musical instruments fashion accessories

Section 4. I believe that categorization as approached in section 4 should precede section 3. The title is misleading. I believe that “Categorization” or sth similar would be more to the point

Parag. 4.2. I think the title “composition” is a more appropriate title than “chemical properties”

Lines 311- 312 and Table 2 should be included in section 2.

Line 365, 368, 373, 383 and elsewhere: Biogas instead of biogasses

Line 368: manufactured??

Line 376-378: Doe the study of Pazera et a refer to the AD of slaughtery by-products for methane production or wastes ?

Line 417: What is meant by Money researchers?

Line 436: The reference of Sanchez et al, is missing from the list

Lines 385-386 and Figure 9: very generic and most probably out of scope since it includes manure

Line 442: …by-products…

Lines 443-447: The meaning is unclear. Please rewrite. Actually the whole pr 5.2.4 is written in a way that is very difficult to follow

Line 466: You have repeated this information (origin or keratin) at leat 3 times!

Lines 465-470: The information is rather generic to be included in par. 5.2.4. Maybe it should be moved to 5.2

Figures 14, 15, 16: All figures are of poor quality. Have the authors crated them or there is an external source, In the latter case the source should be cited! The information of the fig. seems interesting but it is not explained in the text so it actually gets lost! Please either elucidate, or remove the figure

Line 490: Why is Hoshino written with capitals? Nowadays

Line 512: The most common type of protein. Add reference.

Line 513: …mammals, ranging from 25-35%... In developing…

Lines 514-516: please rewrite

Lines 520-526: very generic information. Information on the appropriateness and the processing required so as the collagen derived from slaughterhouse to be used for skin revitalization should be presented , instead.

Line 522: …to  retain… Something is missing here. Please revise.

Parg. 5.3.2. The title should be “Medical applications” or sth similar

Lines 648-672: please include the required information

General comments

  1. Please format the references in a uniform way and according to the requirements of the journal
  2. Editing of the whole MS is required by a native speaker or professional since the language needs extended revision.. Τhere are many grammatical and syntactic errors throughout the text and in many cases the meaning of the sentences is not clear. There are also too many meaningless repetitions of the same information throughout the MS.

Author Response

Reviewer 1

Comments

Reply

General comment

The topic is interesting since the slaughterhouse industry creates indeed several streams of by-products that can be vaporized via different processes, whereas the existing literature mainly deals with the exploitation of wastes and wastewaters. The approach and structure of the MS seem appropriate in general, but the MS seems to have been written in a rush. There are many repetitions of the same information, some information is irrelevant and the use of language is rather poor. The quality of the figures is in general poor, too.

Based on the comments of the reviewer we have checked the entire documents and we have got that repetition of words and information as well. So that the document is updated based on the comments given.

The manuscript is checked by native Kenyan speakers.

Specific comments

The introduction is rather confusing since the differentiation of by-products and actual wastes is not clear. Are the by-products considered as secondary products that can be exploited directly e.g., may be sold to other industrial sectors after separation? Are they discarded contributing to environmental degradation? Is the fate of those by-products different in various parts of the world? I believe these issues should be addressed clearly in the introduction. Moreover, the novelty statement should be emphasized. Which gaps of the existing literature do the study aspires to fill? What is its main aim?

The by produces are taken as secondary products that are discarded and exploited the environments however there is literature that shows some of the by-products are converted into usable materials in the developing countries as well.

For the time being, we have considered mostly the sold by-products.

The discarded by-products are not only affecting the environment but also human and animal health.

As most literature shows the fate of the by-product’s % is different globally because of religion.  The novelty of the manuscript is incorporated into the document

Paragr. 1.1. Overview… I think that in this paragraph you should briefly mention the different sectors of the slaughterhouse industry (meat, poultry) providing numerical and statistical data about the global distribution, handling of products, and general types of by-products. Moreover, here the different streams of by-products generated from the different types of slaughterhouses should be listed as well as the stage at which they are produced.

The required data especially the by-product percentage that is found in slaughterhouses, poultry, and fishery are explained.

Lines 97-107: I believe the water demand is out of scope.

This part of the manuscript is revised

Line 114: please add references

The references are added

Figure 2: I do not think that the scheme is needed. It can be omitted.

The figure is omitted because it hasn’t had additional information

An issue of great importance that is not emphasized in the study is the safety of using slaughterhouse by-products

The safety materials used to dispose of the byproduct are incorporated into the document

Table 1: Please format the table. The fond size and type should be revised.

Table 1: is formatted based on the journal guideline

Figure 3: I think that “Dressed carcass”, should be either “dressed weigh” or “carcass weight” and it is not only the meat. It also includes bones, cartilage etc.

The most common name for the edible part of animal meat is dresses carcass

Line 148: What do you mean by “…the waste is quantified…”

The solid waste percentage of the slaughterhouse are identified

Lines 151-153: Out of scope. Wastewater is not a by-product

Of course, yes, this part is omitted

Line 155: I think that you mean the opposite of what is written here. Landfill is the least desirable option

the landfill one is not encouraged to use because of its high environmental pollution especially in the case of slaughterhouse by-products

Line 170: …by products. Actually…

the by-products are classified as an edible by-products and non-edible by-products

Line 175: are called, instead of we call them

Corrected

Line 180-193: So, landfilling is just throwing away, digging in pits or incineration?

The sentences are revised

Lines 184-185: repetition of the info provided above

 The repetition of words is avoided

Lines 217-219: Repetitions again

Omitted

Paragr. 3.1: The whole paragraph should be rewritten in clearer way. Which by-products can be used for the production of fertilizer? How about its final use? How is sanitation ensured? Manure is certainly not included in the slaughter by-products as presented in Figure 3. Figure 5 does not provide any important information. Please omit

 the paragraphs are restructured and written in a clear way

Lines 242-245: Very generic. Please elucidate

These parts are elaborated on the document

Parag. 3.2. Very generic content.  Which by-products can be used for animal feeds and for which animals? Are there restrictions on their use? How about their sanitation? Fig. 6 is also generic but provides useful info. Please elucidate, providing more detailed information on the steps presented.

Especially the bone and blood by-products are utilized for the production of feed for fish and chicken

Line 248: The meaning is unclear and it sounds a contradiction to the previous paragraph. Please rewrite

The sentences are rewritten

Line 249-250: blood meal is mentioned twice

Repetitions are avoided

Line 251: Do you mean, have to be hydrolyzed to be more easily digested…?

The by-products have to be broken down into small particles for easy digestible by animals

Par. 3.3. Very generic again. Which parts exactly are suitable for glue production and why? The proteins? What are the chemical compounds that are valorised and which are the reagents needed during processing (referring as “ingredients” in Fig. 7)?

The potential use of fats, intestine, bone, and skin proteins are utilized for the production of glue

Figure 7 seems to be incomplete. How are the parts connected? What does the and stand for. Which are the “ingredients”. What is “mechanical filtration”?

 Figure 7 shows that the fats bones and skins are first washed to make clean. And stands for chemical purification and evaporation.  Mechanical filtration means the separation of unusable material from glue

Par. 3.4. I think the title is misleading. Figure 8 has no meaning. Please omit. Line 279: Are musical instruments fashion accessories

 Corrected

Section 4. I believe that categorization, as approached in section 4, should precede section 3. The title is misleading. I believe that “Categorization” or sth similar would be more to the point

The information is rewritten the informal way

Parag. 4.2. I think the title “composition” is a more appropriate title than “chemical properties”

Lines 311- 312 and Table 2 should be included in section 2.

 Ok.

Line 365, 368, 373, 383 and elsewhere: Biogas instead of biogases

Ok.

Line 368: manufactured??

Produced

Line 376-378: Doe the study of Pazera et a refer to the AD of slaughter by-products for methane production or wastes?

Biogas which is produced from the meat processing industry is a well-known method (Pazera et al., 2015).

Line 417: What is meant by Money researchers?

Many scholars

Line 436: The reference of Sanchez et al, is missing from the list.

Incorporated

Lines 385-386 and Figure 9: very generic and most probably out of scope since it includes manure

It shows how the biogas is also produced from the manure of animals

Line 442: …by-products…

hoof, horn, and bones

Lines 443-447: The meaning is unclear. Please rewrite. Actually, the whole pr 5.2.4 is written in a way that is very difficult to follow

The sentences in paragraph 5.2.4 are rewritten

Line 466: You have repeated this information (origin or keratin) at least 3 times!

Corrected

Lines 465-470: The information is rather generic to be included in par. 5.2.4. Maybe it should be moved to 5.2

Ok

Figures 14, 15, 16: All figures are of poor quality. Have the authors crated them or there is an external source, In the latter case the source should be cited! The information of the fig. seems interesting but it is not explained in the text so it actually gets lost! Please either elucidate, or remove the figure

Figures 14, 15, 16 are taken from external sources and are cited

Line 490: Why is Hoshino written with capitals? Nowadays

Corrected

Line 512: The most common type of protein. Add reference.

Reference is added

Line 513: …mammals, ranging from 25-35%... In developing…

Lines 514-516: please rewrite

Rewritten

Lines 520-526: very generic information. Information on the appropriateness and the processing required so as the collagen derived from slaughterhouse to be used for skin revitalization should be presented, instead.

Rewritten

Line 522: …to retain… Something is missing here. Please revise.

To keep

Parg. 5.3.2. The title should be “Medical applications” or sth similar

Corrected

Lines 648-672: please include the required information

The information is updated

Please format the references in a uniform way and according to the requirements of the journal.

The reference is revised based on the guideline of the journal 

Editing of the whole MS is required by a native speaker or professional since the language needs extended revision. There are many grammatical and syntactic errors throughout the text and in many cases the meaning of the sentences is not clear. There are also too many meaningless repetitions of the same information throughout the MS.

The manuscript is checked by native Kenyan speakers.

Reviewer 2 Report

Below I am sending comments regarding the publication:
1. First of all, correct the citations according to the guidelines.
2. Page 1. Line 66. Examples of developing and developed countries can be given.
3. Page 1. Figure 1. Add citation to (FAO, 2019).
4. Do figures 5-16, 19, 20 need no citation?
5. Page 13. Tabel 3. Units should be listed on a separate line.
6. Page 13. Chapter 5.1. Biogas Production. The topic of biogas plants can be expanded to include statistical data from selected countries. In general, it can also be added how we are able to obtain energy from slaughterhouse by-products in other ways.
7. Pages 15-22. Figures 10-13, 15-17 are not referenced in the text.
8. Page 17. Line 427 "Application".
9. Page 18. Line 439 "Diagram"
10. Page 25. Line 615 "Figure".

Author Response

Reviewer 2

Specific comments

First of all, correct the citations according to the guidelines.

The reference is revised based on the guideline of the journal 

Page 1. Line 66. Examples of developing and developed countries can be given.

Examples are given in the manuscript

Page 1. Figure 1. Add citation to (FAO, 2019).

The citation (FAO, 2019) is included in the document

Do figures 5-16, 19, 20 need no citation?

Yes, because the figures are constructed by ourselves, that means we are not taken from someone else

Page 13. Table 3. Units should be listed on a separate line.

The units of by-products are incorporated

Page 13. Chapter 5.1. Biogas Production. The topic of biogas plants can be expanded to include statistical data from selected countries. In general, it can also be added how we are able to obtain energy from slaughterhouse by-products in other ways.

Most scholars explore that the biogas can easily produce from animal manure, organic slaughterhouse by-products, and vegetable biomass together

Pages 15-22. Figures 10-13, 15-17 are not referenced in the text.

The references are incorporated into the document

Page 17. Line 427 "Application".

Utilization instead of application

Page 18. Line 439 "Diagram"

Corrected

Page 25. Line 615 "Figure".

Corrected

Reviewer 3 Report

The article entitled “A Compressive Review on Utilization of Slaughterhouse By-Product: Current Status and Prospect " is a review article in which the authors described the current methods of Slaughterhouse By-Product management.

General note:

The subject of the study is very interesting and topical, with practical importance. Unfortunately, the quality of the manuscript raises many doubts

The introduction is presented in line with the theme, but needs to be improved. It's too long. In the introduction, the authors should only provide an introduction to the topic and the justification for undertaking the research. There should be no charts in the introduction. I recommend that you shorten the Introduction part and remove the figures from that part.

Most of the figures are of poor quality, come from literature, and the authors do not mention the source.

There are some major changes I am suggesting in detailed comments below.

Detailed comments:

The format of the tables needs to be improved, especially Table 1.

In the text, the authors use the wrong format of references to literature. Please check and use the format given in the authors' guidelines. Check and correct throughout the manuscript

Line 57 – there is:percent” – in my opinion there should be “%”. Check and correct throughout the manuscript.

Line 88 – “1.1. Overview of Slaughterhouse Processing Industry” - This part should not be part of the Introduction. It should be a separate manuscript section and should be number 2. Figure 1 should appear in this section

Line 100 – “600-800Litre/animal” – wrong format - Please check and improve format. Check and correct throughout the manuscript.

Line 120 - earlier the authors wrote “by-product”. Please check and improve format. Check and correct throughout the manuscript (e.g. change also in the description of table 1).

Figures are of very bad quality, they need to be corrected (especially Figure 5)

Figure 6 should be removed. Possibly, the quality has to be improved and the source given

Line 290 – there is: ” Physicochemical Properties” what do these properties refer to? please add.

Table 2 is in the part where physical properties should be given. These data, if the authors consider them important, should be included in the earlier part of the article, in which the authors described the amounts of various meat wastes.

Why did the author separate the section "4.1. Physical properties" since they hardly described the physical properties. It needs a lot of improvement.

The format of table 3 must be changed. Redundant column with 1000 kg weight. Authors should use the scientific notation of units e.g. g / kg

Line 364 -  “Biogas Production” is not a perspective, but actual use. That should have been in the earlier section.

Line 375 – “CH4” – wrong format - Check and correct throughout the manuscript.

Figure 10 - the quality has to be improved and the source given

Figures 11, 12 - the quality has to be improved and the source given. Check and correct throughout the manuscript.

Figures 18, 21 - Redundant in my opinion

References

Authors should revise the reference list by following the rules described in the guidelines for authors.

Author Response

                                           Comments

Reply

Reviewer 3

General comment

The subject of the study is very interesting and topical, with practical importance. Unfortunately, the quality of the manuscript raises many doubts

The introduction is presented in line with the theme, but needs to be improved. It's too long. In the introduction, the authors should only provide an introduction to the topic and the justification for undertaking the research. There should be no charts in the introduction. I recommend that you shorten the Introduction part and remove the figures from that part.

Most of the figures are of poor quality, come from literature, and the authors do not mention the source.

There are some major changes I am suggesting in detailed comments below.

We are trying to shorten the introduction part in relation to the title of the manuscript and the manuscript is checked by Kenyan native speakers as well. 

The figures that are taken from kinds of literature are cited but some of the pictures are drawn manually and no need for citation

The figures in the introduction part are removed based on your comments

Specific comments

The format of the tables needs to be improved, especially Table 1.

The format is checked and improved

In the text, the authors use the wrong format of references to literature. Please check and use the format given in the authors' guidelines. Check and correct throughout the manuscript

The reference parts are revised based on the guideline of the journal

Line 57 – there is: „percent” – in my opinion, there should be “%”. Check and correct throughout the manuscript.

The percent is replaced by % throughout the document

Line 88 – “1.1. Overview of Slaughterhouse Processing Industry” - This part should not be part of the Introduction. It should be a separate manuscript section and should be number 2. Figure 1 should appear in this section

Overview of Slaughterhouse Processing Industry are taken as number two

Line 100 – “600-800Litre/animal” – wrong format - Please check and improve format. Check and correct throughout the manuscript.

These parts are revised based on your and reviewer 1 comment

Line 120 - earlier the authors wrote “by-product”. Please check and improve format. Check and correct throughout the manuscript (e.g., change also in the description of table 1).

Corrections are made

Figures are of very bad quality; they need to be corrected (especially Figure 5)

Better quality of figures is incorporated on the document

Figure 6 should be removed. Possibly, the quality has to be improved and the source given

Figure 6 shows how an animal feed is processed from slaughterhouse by-products and the sources are given

Line 290 – there is: ” Physicochemical Properties” what do these properties refer to? please add.

The physical properties and composition of by-products are discussed in detail to determine their utilization area

Table 2 is in the part where physical properties should be given. These data, if the authors consider them important, should be included in the earlier part of the article, in which the authors described the amounts of various meat wastes.

The yield of animal meat processing industry by-product. It shows the detailed percentage of each type of by-product

Why did the author separate the section "4.1? Physical properties" since they hardly described the physical properties. It needs a lot of improvement.

The physical properties and composition of by-products are discussed in detail to determine their utilization area

The format of table 3 must be changed. Redundant column with 1000 kg weight. Authors should use the scientific notation of units e.g. g / kg

The format is changed and the units are corrected

Line 364 -  “Biogas Production” is not a perspective, but actual use. That should have been in the earlier section.

In the developing countries especially in Ethiopia biogas production is not an actual production process rather it is the prospect

Line 375 – “CH4” – wrong format - Check and correct throughout the manuscript.

Corrected

Figure 10 - the quality has to be improved and the source given

The quality of the figures is revised and the sources are incorporated 

Figures 11, 12 - the quality has to be improved and the source given. Check and correct throughout the manuscript.

The quality of the figures is revised and the sources are incorporated 

Figures 18, 21 - Redundant in my opinion

Figure 18 discussed Collagens used for bone graft substitute and dressing and powder and Figure 21 is the Utilization of the intestine for surgical sutures.

Authors should revise the reference list by following the rules described in the guidelines for authors.

 The reference parts are revised based on the guideline of the journal

Round 2

Reviewer 1 Report

I am afraid that the revision is insufficient. The response to comments is and the changes made are not adequate and the MS was hardly edited, overall. The language remains very poor with several, grammatical, syntactic, and conceptual errors. Some examples and possible corrections are given below:

Lines 31-34: Too long sentence and has many grammatical and syntax errors. (as a substitute, health issue, depletion of space, which was expense) Please revise

Lines 34-35:What is the meaning of “ The environmental; uncomfortable disposal of is .”?

Lines 36-37: “An option to address the environmental impact and health problems on the population, especially that living around…”

Line 38: Omit by-product

Line 38: “..is the conversion of abattoirs wastes to biomateriasl, biofuels, biogas and biochemicals as,  well.”

Line 40: “ the by-products of the slaughterhouse will be utilized  appropriately, contributing  not only to the  reduction of environmental pollution (contamination) but also to the creation of value, attracting thus investors in the  the meat industry”

Line 43: “  are generated  during the processing  of livestock for meat production, and are …”

Line 48: “…however, conversion of by-produces to biomaterials is of great importance to contribute to a green economy…”

Lines 50-51: “are an energy-rich materials  consists of collagen….nd minerals.”

The Figures are of poor quality, too.

The reference list is not uniform and not formatted according to the requirements of the journal (see below)

References should be described as follows, depending on the type of work:

  • Journal Articles:
    1. Author 1, A.B.; Author 2, C.D. Title of the article. Abbreviated Journal Name Year, Volume, page range.
  • Books and Book Chapters:
    2. Author 1, A.; Author 2, B. Book Title, 3rd ed.; Publisher: Publisher Location, Country, Year; pp. 154–196.
    3. Author 1, A.; Author 2, B. Title of the chapter. In Book Title, 2nd ed.; Editor 1, A., Editor 2, B., Eds.; Publisher: Publisher Location, Country, Year; Volume 3, pp. 154–196.

Since the topic is very interesting and the structure adequate in general, please proceed with a more careful revision.

Author Response

S/n

Comments

Reply

Reviewer 1

I am afraid that the revision is insufficient. The response to comments is and the changes made are not adequate and the MS was hardly edited, overall. The language remains very poor with several, grammatical, syntactic, and conceptual errors. Some examples and possible corrections are given below:

I have tried a lot to make it better the whole word document. The overall document is reviewed by native speakers and almost all grammatical, syntactic and conceptual errors are corrected

Lines 31-34: Too long sentence and has many grammatical and syntax errors. (as a substitute, health issue, depletion of space, which was expense) Please revise

Line 31-34 are rewritten and explained in a better way

Lines 34-35: What is the meaning of “ The environmental; uncomfortable disposal of is .”?

The sentence is restructured

Lines 36-37: “An option to address the environmental impact and health problems on the population, especially that living around…”

The superlative option to handle those environmental and health problems using an integrated biorefinery technique for the conversion of slaughterhouses waste to biomaterial, biofuel, biogas, and biochemicals

Line 38: Omit by-product

Corrected

Line 38: “..is the conversion of abattoirs wastes to biomaterials, biofuels, biogas and biochemicals as, well.”

The sentence is restructured

Line 40: “the by-products of the slaughterhouse will be utilized appropriately, contributing not only to the reduction of environmental pollution (contamination) but also to the creation of value, attracting thus investors in the   meat industry”

The sentence is restructured

Line 43: “  are generated during the processing of livestock for meat production, and are …”

The paragraph as a whole is rewritten  

Line 48: “…however, conversion of by-produces to biomaterials is of great importance to contribute to a green economy…”

Worldwide the slaughterhouse industries produce a large amount of organic by-products, however; conversion of by-mass to usable biomaterials has great importance on facilitating a green economy, consistent supplying of biomaterials for the industries

Lines 50-51: “are an energy-rich materials consists of collagen….and minerals.”

Slaughterhouse by-products are protein-rich biomass consisting of collagen (gelatine), keratin, fats, amino acids, and mineral products.

The Figures are of poor quality, too.

The figures are almost changed

The reference list is not uniform and not formatted according to the requirements of the journal (see below)

 Corrected

Reviewer 2 Report

In principle my comments were taken into account, but the review of energy production could be further developed.

Author Response

S/n

Comments

Reply

Reviewer 2

In principle, my comments were taken into account, but the review of energy production could be further developed.

Based on the reviewer’s comment the energy production part is developed and restructured. Check on the document the details

Reviewer 3 Report

I appreciate the author efforts on this manuscript, which indeed improve the quality of this manuscript. Particularly,  the authors added missing information. Thus, I satisfy the authors' respondence and the revision.

Author Response

S/n

Comments

Reply

Reviewer 3

I appreciate the author’s efforts on this manuscript, which indeed improve the quality of this manuscript. Particularly, the authors added missing information. Thus, I satisfy the authors' respondence and the revision.

Ok.

Round 3

Reviewer 1 Report

I understand the authors tried to do their best, but the revision is still unsufficient! The use of english is very problematic; the text is difficult for the readers to follow and in many points the sentences make  no sence.

E.g in the revised text in the very beggining of the introduction (line 34) it is stated "To reduce the environmental problem due to landfills of by-products, it
should take into consideration on the production of fertilizer and animal feed and bio- materials is the effective utilization of slaughterhouse wastes as fertilizer and animal feed." The sentence has noumeros errors, whereas the meaning is not any clear at alI. I am afraid that the language is not acceptable for  a written document of any type, even more for a scientific publication.

The figures are of low quality and most of them are not even needed, since they are not exaplanatory, or summurize graphically, or provide any insight on the discussed topic of the main text.

The structure and the approach of the study are in general sufficient, but the MS cannot ve published in its curremt form.